# Increased Levels of Beta-Endorphin and Noradrenaline after a Brief High-Impact Multidimensional Rehabilitation Program in Multiple Sclerosis

**DOI:** 10.3390/life12050755

**Published:** 2022-05-19

**Authors:** Alessia d’Arma, Marina Saresella, Valentina Rossi, Ivana Marventano, Federica Piancone, Francesca La Rosa, Mario Clerici, Laura Mendozzi

**Affiliations:** 1IRCCS Fondazione Don Carlo Gnocchi, Via Capecelatro 66, 20148 Milan, Italy; adarma@dongnocchi.it (A.d.); nutrizione@vrossi.it (V.R.); imarventano@dongnocchi.it (I.M.); fpiancone@dongnocchi.it (F.P.); flarosa@dongnocchi.it (F.L.R.); mario.clerici@unimi.it (M.C.); lmendozzi@dongnocchi.it (L.M.); 2Department of Pathophysiology and Transplantation, Faculty of Medicine and Surgery, University of Milan, Via Sforza 35, 20122 Milan, Italy

**Keywords:** multiple sclerosis, lifestyle, rehabilitation, beta-endorphin, catecholamines, immune system, neuroendocrine system

## Abstract

Finding new solutions for the management of multiple sclerosis (MS) is crucial: further research is needed to study the effect of non-pharmacological interventions on the symptoms and the course of the disease, especially on lifestyle. Benefits from a proper lifestyle are evident not only on a clinical level but also on immune and neuro-endocrine systems. A brief high-impact multidimensional rehabilitation program (b-HIPE) was proposed for a sample of people with MS (pwMS) with a medium level of disease disability. We tested the change on clinical parameters and quality of life (QoL) after participation in B-HIPE. We furthermore decided to measure beta-endorphin and catecholamines concentrations pre- and post-participation in the b-HIPE program, due to the relationship between these hormones and the immune system in neurodegenerative diseases. Our results showed that after the b-HIPE program, an improvement of clinical parameters and QoL occurred. Moreover, we found higher levels of beta-endorphin and noradrenaline after participation in the program. These findings highlight the importance of implementing lifestyle interventions in the clinical management of MS. Furthermore, we hypothesize that the B-HIPE program increased beta-endorphin and noradrenaline levels, helping to reduce the inflammation related to MS disease.

## 1. Introduction

Multiple sclerosis (MS) is a neurodegenerative disease of the central nervous system (CNS) affecting young adults. Due to this fact and its unpredictable prognosis, in addition to the physical impairment, MS can affect patient QoL and daily living functioning. This aspect, together with the lack of definitive cures, enhances the need for lifestyle intervention [1,2,3,4], to go alongside the several pharmacological treatments available for MS, whose aim is reducing the number and the severity of the relapses, preventing or slowing the progression of disability [5]. In essence, encouraging a proper lifestyle (healthy eating, routine physical activity or exercise, and avoiding smoking) should be a fundamental message to pwMS [6].

We recently published a scientific work in which a multidimensional rehabilitation program focused on lifestyle showed an improvement not only on clinical parameters but also in the immunological parameters of these patients [7,8,9].

The immunological system is connected with the neuro-endocrine system [10,11] by the hypothalamus-pituitary-adrenal (HPA) axis involvement in the integration of the body’s stress response with immune activity [12,13]. Notably, neurotransmitters, neuropeptides, and hormones such as catecholamines (adrenaline, noradrenaline, and dopamine), β-endorphins, serotonin, and cortisol, can activate immune cells and chemokines, and cytokines produced by immune cells can regulate the stress response along the HPA axis [13,14,15]. The result of this bi-directional communication is to maintain homeostasis. Behavioral interventions targeted at alleviating stress, such as physical exercise, have been indicated to reduce inflammatory parameters by neuropeptide factor involvement [16,17].

In this perspective, studying the interactions between MS and the neuroendocrine-immune system may help us comprehend the functioning of new therapeutic approaches for the treatment of MS.

This study aimed to test the B-HIPE program efficacy on clinical parameters and the quality of life (QoL) on pwMS, with a specific focus on the impact on the neuroendocrine-immune mechanisms. 

## 2. Materials and Method

### 2.1. Participant Enrollment

A total of 15 subjects with MS (pwMS) were consecutively recruited from the Multiple Sclerosis Center—Neuromotor Rehabilitation Unit of Don Carlo Gnocchi Foundation, IRCCS, in Milan (Italy). 

*Inclusion criteria*. Inclusion criteria were as follows: (1) diagnosis of relapsing–remitting (RR) and secondary progressive (SP) MS, (2) age ≥ 18 and ≤70 years, (3) no change of pharmacological disease-modifying treatments in the six months before enrolment, (4) no clinical relapse or use of steroid treatment in the three months before enrolment, (5) pwMS who were on a Western diet, (6) pwMS with motor control of upper limbs sufficient to maneuverer a tiller, and (7) informed consent provided for study participation. 

*Exclusion criteria*. Exclusion criteria were as follows: (1) history of nervous system disorders other than MS; (2) unstable psychiatric illness, such as psychosis or major depression; (3) severe disability according to the Expanded Disability Status Scale (EDSS) score > 8; (4) severe cognitive impairment (i.e., dementia), according to the patient’s medical records; (5) severe visual impairment; (6) alcohol and drug abuse; (7) and dysphagia and/or comorbidities requiring protected environments and specific medical assistance. 

All pwMS enrolled in this study found themselves in a critical stage of their disease in terms of lack of motivation and critical issues associated with their disease: the B-HIPE program aims to help pwMS to dismiss the passive role of “patient”, gaining the “active role” of “participant” in their own pathway to care.

The study was conducted in compliance with the Helsinki Declaration of 1975, as revised in 2008. A local ethics committee approved the study (number 07_23/05/2018), and written informed consent to be included in the study was obtained from participants before the study’s initiation.

### 2.2. Procedure of the Study

Participants participated in a brief high-impact preparatory experience (B-HIPE) described in our previous paper [18].

The B-HIPE program is a multidimensional rehabilitation program focused on the change of bad habits in PwMS. 

Several activities are integrated:(a).Neuromotor rehabilitation.(b).Recommended diet based on the Mediterranean diet principles: fresh fruits and vegetables, whole grain products, legumes, nuts and seeds, fish, eggs, and a small amount of poultry and dairy products. Red meat, processed meat, and alcoholic and sweet drinks are excluded.(c).Sailing course designed to accommodate disabled sailors.(d).Mindfulness with participation extended to all staff members.

### 2.3. Assessment of Clinical and Qualitative Variables

After the enrollment, all the participants were evaluated through a clinical assessment administered by the neurologist. 

Subsequently, at baseline (T0) and after intervention (T1), clinical and quality of life (QoL) questionnaires were administered:(a).SF-36 Health-Related Quality of Life questionnaire (HRQoL), a measure for the Health-Related Quality of Life;(b).Hospital Anxiety and Depression Scale (HADS), a self-assessment scale for detecting states of depression and anxiety;(c).The Medical Outcomes Study Sleep Scale (MOSS), a scale evaluating sleep disturbance and sleep adequacy, with a sleep problems index;(d).Epworth Sleepiness Scale, a scale for measuring daytime sleepiness;(e).International Restless Legs Syndrome Scale (IRLSS), a scale evaluating restless leg syndrome severity.

### 2.4. Blood Sample Collection

To study neuro-endocrine parameters, whole blood was collected in vacutainer tubes (Becton Dickinson and Co., Rutherford, NJ, USA) at baseline (T0) and after intervention (T1). The blood samples were centrifuged at 3000 rpm for 10 min to separate sera. Serum was stored at −80 °C until use.

Plasma was obtained from 10 milliliters of whole blood collected in EDTA-containing vacutainer tubes (Becton Dickinson and Co., Rutherford, NJ, USA), centrifuged at 3000 rpm for 10 min, and stored at −80 °C until use.

#### 2.4.1. Cortisol Detection

Serum concentrations of cortisol were determined by competitive enzyme immunoassay using a Cortisol Parameter Assay Kit (R&D Systems, Inc.614 McKinley Place NE Minneapolis, Minneapolis, MN, USA) according to the manufacturer’s recommendations. The optical densities (OD) were determined at 450/620 nm. Sensitivity was 0.111 ng/mL; assay range was 0.2–10 ng/mL.

#### 2.4.2. β-Endorphin Detection

Serum concentrations of β-endorphins were determined by competitive enzyme immunoassay using a Human Beta Endorphin ELISA Kit (R&D Systems) according to the manufacturer’s recommendations. The OD were determined at 450 nm. Sensitivity was 5.11 pg/mL; assay range was 12.35–1000 pg/mL. 

#### 2.4.3. Serotonin Detection

Serum concentrations of serotonin were determined by competitive enzyme immunoassay using a Human Serotonin ELISA Kit (R&D Systems) according to the manufacturer’s recommendations. The OD were determined at 450 nm. Sensitivity was <50 ng/mL, assay range was 50–1000 ng/mL. 

#### 2.4.4. Catecholamine Detection

Plasma concentrations of adrenaline, noradrenaline, and dopamine were determined by ELISA immunoassay using a Human TriCAT TM ELISA Kit (IBL International–TECAN Group, Männedorf, Switzerland) according to the manufacturer’s recommendations. The OD were determined at 405/620 nm. Sensitivity (S) and assay range (AR): adrenaline: S = 8 pg/mL, AR = 1.5–150 ng/mL; noradrenaline: S = 20 pg/mL, AR = 5.0–500 ng/mL; dopamine: S = 4 pg/mL, AR = 6–11,470 ng/mL

A plate reader (Sunrise, Tecan, Mannedorf, Switzerland) was used for ELISA immunoassay.

### 2.5. Statistical Analyses

The analyses were performed with the Jamovi (The jamovi project (2021). jamovi (Version 1.6) [Computer Software] Sydney, Australia. Retrieved from https://www.jamovi.org) and SPSS (Version 24.0, IBM, Milan, Italy) software.

Descriptive analyses were performed for demographic characteristics, including absolute counts and percentages for categorical variables, means, standard deviation (SD), medians, and range for continuous variables as appropriate. 

To test if and how after our B-HIPE program clinical and neuro-immuno-endocrine variables could change, a statistical comparison with Wilcoxon test between pre vs. post intervention was performed.

Differences in neuroendocrine parameters were assessed by the Wilcoxon test using the MEDCALC statistical package (MedCalc Software bvba, Mariakerke, Belgium).

## 3. Results

### 3.1. Results of Demographic Characteristics and Clinical Variables

We collected data from 15 pwMS—eight males and seven females. Among these, 10 had an RR MS diagnosis and 5 an SP MS diagnosis. Principal descriptive demographic characteristics are shown in Table 1.

In Table 2, we report the clinical results of the Brief High-Impact Preparatory Experience (B-HIPE) program. In all clinical variables, we found a statistically significant improvement, except for the HADS scale—Depression part-, IRLSS scale, and two subdomains of the SF-36, the Physical Functioning and General Health. However, these latter variables reached a tendency towards statistical significance.

### 3.2. HPA Axis Factors

The neuroendocrine factors adrenaline, noradrenaline, dopamine, β-endorphins, cortisol, and serotonin were analyzed in sample sera of MS patients before (T0) and after the rehabilitation program (T1) by an ELISA kit (Table 3).

The results showed an increase in β-endorphins (113 pg/mL) and noradrenaline (560 pg/mL) in T1 of the MS patient group compared to T0 (β-endorphins: 142 pg/mL; noradrenaline: 606 pg/mL; *p* < 0,02 for both); these data are shown in Figure 1.

## 4. Discussion

The pwMS enrolled in our study participated in B-HIPE, an intervention focused on the disposal of a passive attitude and the promotion of self-efficacy, which are very helpful in changing lifestyle. Physiotherapy, mindfulness, sailing, healthy eating, and cultural activities were experienced in a leisure environment of a seaside village at La Maddalena (Sardinia, Italy).

Herein, we tested its influence on the clinical, qualitative parameters and neuroendocrine mechanisms of pwMS. 

Our results showed that B-HIPE is able to modify both clinical and neuroendocrine systems. 

On the basis of clinical and qualitative results, our B-HIPE program impacted the QoL and the quality of sleep, with a statistically significant difference pre- vs. post-intervention in these variables evaluated through the SF-36 and MOSS/ESS scales. These results are particularly relevant, considering that the literature reports a lower health-related quality of life (HRQoL) of pwMS than general and other chronic disease populations [19]. Furthermore, moderate or severe sleep problems are widespread in MS [20], higher than that of chronically ill subjects and the general population [21]. 

As observed in the literature, all these factors are strictly related: in fact, alterations in natural sleep could result in adverse changes, including fatigue [22], anxiety and depression [23], and decreased quality of life [24]. Regarding anxiety status, pwMS included in our study did not show pathological levels at baseline; however, our work showed that the B-HIPE program can further reduce these traits, opening up the possibility of use with people who present these types of symptoms. We also found a significant reduction in the “pain” subscale of SF-36. Pain is also related to sleep disorders [25]. 

The B-HIPE can also impact subjective energy and fatigue: the results at the energy/fatigue subscale of SF-36 indicated a subjective improvement of these features after participation to the B-HIPE. 

In addition to the improvement detected in QoL parameters, we found a statistically significant increase in noradrenaline and β-endorphins levels. The improvement detected in almost all the qualitative scales could be related to the significant increase in noradrenaline and β-endorphin levels, suggesting a positive effect of the B-HIPE on these neuroendocrine factors.

Overall, the improvement detected in almost all the qualitative scales could be related to the increase in noradrenaline and β-endorphin levels because recent literature has shown correlations between the HPA axis and the body’s stress response. Exercise increases levels of circulating β-endorphins and reduces pain, playing a significant role in pain relief [26]. 

Moreover, Patel and colleagues [27] highlighted the correlation between elevated β-endorphin levels and QoL scores, especially for the physical and social functioning roles, suggesting that β-endorphins could be potential biomarkers for physical disease status in MS because of its anti-inflammatory properties. 

High noradrenaline levels have been found after participation in the B-HIPE, suggesting the involvement of this neuroendocrine factor in ameliorating clinical status in MS. Notably, noradrenaline reduces inflammatory responses and increases the expression of several neurotrophins, including BDNF [28], suggesting the possible use of noradrenaline-based approaches as an alternative or additional therapies in MS. The increase in noradrenaline levels through administration of drugs ameliorated the MS course, with experienced substantial improvements in sensory, motor, and autonomic symptoms confirming the beneficial role of this neuroendocrine factor in MS [29,30]. The data here indicate a close association among rehabilitation programs, neuroendocrine factors, and anti-inflammatory and neuroprotective abilities, suggesting a possible role of the latter in new therapeutic and rehabilitative strategy treatment.

The main limits of our study are intrinsic to the design of our B-HIPE program. Because we desired the realization of a very comprehensive rehabilitation program with a specific focus on proper diet and mental well-being, it is not possible to enrollment more than 5/6 participants for each edition: safety of the participants was the most important factor. This solution would not allow, however, the access to a large cohort of patients to our service, and thus the generalization of our program is still limited. Future directions will regard the possibility of extending this program to a larger population with a controlled trial study design in order to verify the impact of a multidimensional rehabilitation on clinical and catecholamines parameters.

In conclusion, our study reinforces the hypothesis that a multidimensional intervention focused on a proper lifestyle could represent a good proposal in the array of interventions dedicated to the management of the progression of this challenging chronic disease, due to the well-known relationship between lifestyle and the reduction of the inflammatory process in MS pathology [31]. Lifestyle factors (including physical activity, mental well-being, and an anti-inflammatory diet) should be increasingly considered in the field of rehabilitation, since chronic diseases, especially multiple sclerosis, are now viewed as “lifestyle diseases” [32].

## Figures and Tables

**Figure 1 life-12-00755-f001:**
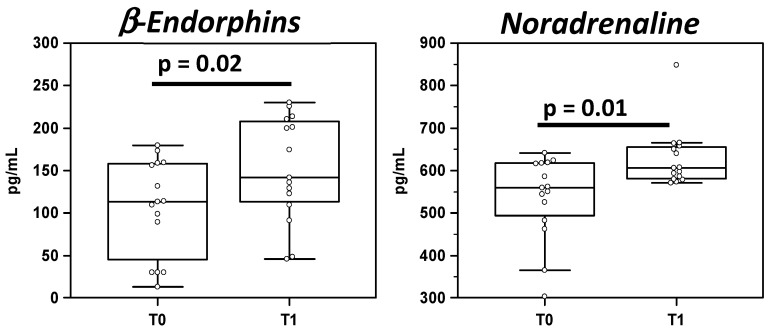
β-Endorphins and noradrenaline concentration (pg/mL) detected by ELISA in serum sample of MS patients at baseline (T0) and at the end of the B-HIPE (T1). The statistical analyses were assessed by Wilcoxon test using the MEDCALC statistical package. Summary results are shown in the bar graphs. The dots represent all data; the boxes stretch from the 25th to the 75th percentile; the line across the boxes indicates the median values; the lines stretching from the boxes indicate extreme values. Significant differences are shown.

**Table 1 life-12-00755-t001:** Demographic characteristics of the enrolled pwMS.

Demographic Characteristics of pwMS	
** *N* **	15
** *Sex* ** *(M/F)*	8/7
** *Age* ** *(years) [average ± SD (min–max)]*	49.13 ± 8.52 (36–69)
** *MS course* ** *(RR/SP)*	10/5
** *Disease duration* ** *(years) [average ± SD (min–max)]*	19.38 ± 5.05 (7–28)
** *EDSS score* ** *[average ± SD (min–max)]*	5.4 ± 1.66 (2–8)

Abbreviations: N = number; M = male; F = female; MS = multiple sclerosis; Min = minimum; Max = maximum; RR = relapsing–remitting; SP = secondary progressive; EDSS = Expanded Disability Status Scale.

**Table 2 life-12-00755-t002:** Results at clinical variables before and after the B-HIPE program.

Variable	Time	Mean ± SD	*p*-Value
MOSS *	T0	25.1 ± 14.2	**0.023**
	T1	15.0 ± 8.8	
ESS *	T0	6.73 ± 3.97	**0.018**
	T1	3.93 ± 3.26	
IRLSS *	T0	13.8 ± 9.59	0.080
	T1	10.7 ± 7.15	
HADS anxiety *	T0	4.07 ± 4.20	**0.014**
	T1	2.67 ± 3.81	
HADS depression *	T0	4.40 ± 4.53	0.408
	T1	3.40 ± 4.07	
SF-36 physical functioning #	T0	463 ± 303	0.063
	T1	560 ± 345	
SF-36 role limitation due to physical health #	T0	163 ± 142	**0.004**
	T1	333 ± 129	
SF-36 role limitation due to emotional problems #	T0	180 ± 126	**0.013**
	T1	287 ± 35.2	
SF-36 energy/fatigue #	T0	208 ± 53.3	**0.002**
	T1	300 ± 66.3	
SF-36 emotional wellbeing #	T0	340 ± 99.1	**0.003**
	T1	439 ± 76.1	
SF-36 social functioning #	T0	116 ± 47.8	**0.004**
	T1	153 ± 44.2	
SF-36 pain #	T0	146 ± 49.0	**0.004**
	T1	185 ± 32.3	
SF-36 general health #	T0	250 ± 106	0.064
	T1	305 ± 125	

Abbreviations: MOSS = Medical Outcomes Sleep Scale; HADS = Hospital Anxiety and Depression Scale; ESS = Epworth Sleepiness Scale; IRLSS = International Restless Legs Syndrome Scale; SF-36 = 36-Item Short Form Survey. * = lower values indicate an improvement. # = higher values indicate an improvement.

**Table 3 life-12-00755-t003:** β-endorphins, adrenaline, dopamine, noradrenaline, cortisol, and serotonin concentration (pg/mL) detected by ELISA in serum sample of MS patients at baseline (T0) and at the end of the B-HIPE (T1). The statistical analyses were assessed by Wilcoxon test using the MEDCALC statistical package. Median, interquartile range, and significant differences are shown.

HPA Axis Factors (pg/mL)	Median	Interquartile Range	*p*-Value
β-Endorphins T0	**113.4**	**45.2–158.3**	***p* = 0.02**
β-Endorphins T1	**142.0**	**113.2–208.3**	
Adrenaline T0	402.0	336,750–432.0	n.s.
Adrenaline T1	392.0	376,750–411.3	
Dopamine T0	21,522.0	18,208.0–24,376.0	n.s.
Dopamine T1	21,588.0	20,308.0–23,047.8	
Noradrenaline T0	**560.0**	**493.8–617.5**	***p* = 0.02**
Noradrenaline T1	**606.0**	**581.5–656.0**	
Cortisol T0	165.3	140.3–202.7	n.s.
Cortisol T1	180.6	114.2–188.8	
Serotonin T0	162.5	76.6–406.3	n.s.
Serotonin T1	104.0	69.2–441.3	

Abbreviations: n.s. = no significant.

## Data Availability

The datasets used and/or analyzed during the current study are available from the corresponding author upon reasonable request.

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
