# Peer review of "Increased Levels of Beta-Endorphin and Noradrenaline after a Brief High-Impact Multidimensional Rehabilitation Program in Multiple Sclerosis"

_life, 2022, doi:10.3390/life12050755_

Round 1

Reviewer 1 Report

The Authors examined the impact of the Brief High Impact Preparatory Experience (B-HIPE) program, as a part of the LINUS project, on the overall neurological status of the subjects and on serum levels of chosen neuroendocrine factors. This study is potentially interesting as changes in daily habits and overall quality of life of MS patients may have long-term beneficial effects on disease progression, especially when combined with proper medications. 

However, there are a few concerns that the Authors need to address:

1) It is unclear to this Reviewer which statistical test was performed in this study. Since there is a pre vs. post-treatment comparison, paired t-test should be appropriate. Please be more specific.

2) The Authors should explain why they mixed RRMS patients and SPMS patients. These two stages are quite different in terms of the underlying pathogenesis and severity of the disease. 

3) Graphics depicted in Figure 1 should be more organized and with individual values. Figure legends should be more precise and state the statistical analysis that was performed.

4) Line 53-55 should be rephrased since the methods that were used in this study could not be used to reveal the mechanism of the B-HIPE program, but rather to detect simple post festum changes. 

5) Did the Authors detect changes in EDSS score following the B-HIPE program?

6) How do Authors explain no changes in other factors examined (i.e. serotonine, dopamine etc.)

7) Overall English needs more editing and many sentences should be rephrased. 

Author Response

Dear reviewer

We carefully read the critiques and have modified the manuscript according to your critiques. All the changes in the revised version of the manuscript are highlighted in yellow.

We feel that the manuscript is now leaner and stronger; we hope you will agree with us and that this version will be adequate for the qualitative standards of   Life.

Best regards

Marina Saresella

Reviewer 2 Report

This is a very small sample with no control. The intervention seems to have been very relaxing and enjoyable, so it does not surprise me that QOL measures and endorphins were improved after a pleasant time in the Mediterranean. As such, I'm reluctant to accept that these results are the result of the intervention itself, and not just the result of having a refreshing vacation. I think that this work is unfortunately not strong enough for publication in a peer-reviewed journal. Perhaps a poster presentation at a conference may be a better avenue for these findings.

Other thoughts: 

You mention that the Wilcoxon test was used for the neuroendocrine parameters. Was this also how clinical parameters were assessed?

In Table 1, be consistent in using either commas or points to indicate decimals.

In Table 2, it would be helpful to indicate which direction indicates improvement for each of these measures. 

SF-36 subscale scores should be calculated by averaging the questions included in each subscale, not adding. So, each subscale score should range between 0-100. 

Figure 1 (boxplots) looks suspicious to me. I find it hard to believe that you got statistically significant results from such a small sample, and with so much overlap of the inter-quartile ranges. Including your test statistic may be helpful. 

Table 3: I would report the median and interquartile range of the significant results here as well, perhaps bolding them.

Author Response

(The authors gave the same response as above.)

Reviewer 3 Report

One of the inclusion criteria of the study was pwMS following a Mediterranean diet. How was it possible to know that this diet was followed by the patients?

There is no brief explanation of the psychometric characteristics of the tests.

The sample collection procedure (randomization?) is not made explicit. A comment on the representativeness of the sample should be added: why authors have not chosen a larger number of participants because of possible losses during the study?

It is not clear how long the study lasted and whether there was any loss of patients during the study.

the limitations of the study should be specified

the authors should add a paragraph as the final conclusion of the study with the main ideas

Author Response

(The authors gave the same response as above.)

Round 2

Reviewer 1 Report

The Authors addressed all concerns raised by this reviewer.

Author Response

Dear Reviewer,

I sincerely thank you, as your remarks allowed us to critically improve the results.

sincerely

Marina Saresella 

Reviewer 2 Report

Improvements have been made, but I simply don't think this study is compelling or warrants a full-length article, for the reasons given previously. 

Author Response

(The authors gave the same response as above.)

Reviewer 3 Report

After further revision of the text, the appropriate changes and clarifications have been made.
Congratulations 

Author Response

Dear Reviewer,

I sincerely thank you, as your remarks allowed us to critically improve the results.

sincerely

Marina Saresella 

This manuscript is a resubmission of an earlier submission. The following is a list of the peer review reports and author responses from that submission.